# The Alpha Variant (B.1.1.7) of SARS-CoV-2 in Children: First Experience from 3544 Nucleic Acid Amplification Tests in a Cohort of Children in Germany

**DOI:** 10.3390/v13081600

**Published:** 2021-08-12

**Authors:** Meike Meyer, Anna Holfter, Esra Ruebsteck, Henning Gruell, Felix Dewald, Robert Walter Koerner, Florian Klein, Clara Lehmann, Christoph Huenseler, Lutz Thorsten Weber

**Affiliations:** 1Department of Pediatrics, Faculty of Medicine and University Hospital Cologne, University of Cologne, 50937 Cologne, Germany; esra.ruebsteck@uk-koeln.de (E.R.); robert.koerner@uk-koeln.de (R.W.K.); christoph.huenseler@uk-koeln.de (C.H.); lutz.weber@uk-koeln.de (L.T.W.); 2Departement of Pediatrics, Elisabeth-Krankenhaus Rheydt, 41239 Moenchengladbach, Germany; anna.holfter@sk-mg.de; 3Institute of Virology, Faculty of Medicine and University Hospital Cologne, University of Cologne, 50937 Cologne, Germany; henning.gruell@uk-koeln.de (H.G.); felix.dewald@uk-koeln.de (F.D.); florian.klein@uk-koeln.de (F.K.); 4Department I of Internal Medicine, Faculty of Medicine and University Hospital Cologne, University of Cologne, 50937 Cologne, Germany; clara.lehmann@uk-koeln.de; 5German Center for Infection Research (DZIF), 38124 Braunschweig, Germany

**Keywords:** Germany, COVID-19, Alpha, children, Ct-value

## Abstract

In May 2021, the Alpha variant (B.1.1.7) of SARS-CoV-2 was found in 91% of the SARS-CoV-2 cases in Germany. Not much is known about the symptoms, courses of disease, and infectiousness in pediatric patients with the Alpha variant. Objective: The aim of this retrospective analysis was to gain information on the infection with the Alpha variant in children and adolescents. Methods: Between 12 January 2021 and 3 June 2021, all nucleic acid amplification tests (NAATs) of children who received a swab for SARS-CoV-2 were included. Data were collected on standardized questionnaires. The analysis of data was anonymized and retrospective. Results: We investigated 3544 NAATs; 95 children were tested positive (2.7%) for SARS-CoV-2. For the sub-analysis, 65 children were analyzed. In 59 children, the Alpha variant was found (90.8%), and 54.2% (*n* = 32/59) were symptomatic. The most common symptoms were fever, cough, and rhinitis. The median Ct value was 24.0 (min 17.0; max 32.7). Conclusions: We can underline early findings that children are still less effected by SARS-CoV-2 infection with the spread of the Alpha variant. We found no evidence that children infected with the Alpha variant showed more severe symptoms or suffered from a more severe clinical course than those infected with the wild type.

## 1. Introduction

Since the occurrence of the first known case of SARS-CoV-2 in China in December 2019, the pandemic has spread all over the world. As per June 2021, there have been 173 million documented cases worldwide, affecting all age groups [1]. In Germany, almost 3.7 million verified cases of SARS-CoV-2 have been recorded since the first report of a German patient at the end of January 2020 [2]. Of these cases, 581,399 were 0–19 years old [3].

From early on, there have been findings that the course of the resulting coronavirus disease (COVID-19) is less severe in children compared with adults [4]. A recent statement by the DGPI (German Society for Pediatric Infectiology) highlights the extreme rarity of a severe or fatal outcome after SARS-CoV-2 infection in children [5]. Nevertheless, the question of the risks for children for infection and transmission keeps on engaging pediatricians, occupying politics and worrying families. New factors like the spread of viral mutants and more reports of severe secondary diseases like MIS-C (multisystem inflammatory syndrome in children) need to be counted into the equation for understanding the impacts of COVID-19 on children and adolescents.

Mutations in the gene encoding the viral spike protein have resulted in the development of numerous SARS-CoV-2 variants. In mid-December 2020, the first reports from the United Kingdom described a rapidly spreading viral variant that was later designated as the Alpha variant (B.1.1.7) of SARS-CoV-2. Shortly after, this variant emerged in other countries, including Germany [6]. By May 2021, it was the cause of more than 90% of SARS-CoV-2 infections in Germany [7]. While the increased transmissibility of the Alpha variant compared to previously circulating variants has been well established, there is inconclusive evidence on disease severity of infections with the Alpha variant compared to other variants, particularly in children and adolescents [8,9].

This work presents data gathered over a period of 6 months from January to June 2021 from 3544 SARS-CoV-2 nucleic acid amplification tests (NAATs) in children and adolescents between the age of 0–18 years. The aim is to provide information on how the rising numbers of the Alpha variant may have affected the symptoms, courses of disease (e.g., hospital admission rates), and viral loads in children as reflected by the PCR cycle threshold (Ct) values.

## 2. Materials and Methods

We included all SARS-CoV-2-NAATs from children and adolescents between the age of 0–18 years who received a naso- and/or oro-pharyngeal swab between 12 January 2021, when the routine variant mutational analysis started, and 3 June 2021 in this analysis. The swabs and analysis were carried out at the University Hospital of Cologne, Germany. All patients were tested upon admission, regardless of whether they showed symptoms for SARS-CoV-2 or not. In addition, outpatients with symptoms or exposure to SARS-CoV-2-infected individuals, as well as patients receiving outpatient clinic care, underwent a swab for SARS-CoV-2 NAAT. We cannot preclude the possibility that some patients were tested repeatedly within the observation period, as patients with an underlying chronic disease often regularly visit the hospital. However, for our analysis, we regarded the number of tests performed to be of consequence rather than the number of tested patients. For the mutational sub-analysis, samples with Ct values ≥ 33 were excluded, because variant analysis was only performed when Ct-values were <33. Furthermore, we excluded patients where no Ct value was available or no analysis of variants was performed for whatever reason. For more information, see Figure 1.

All swabs were conducted by trained staff and were tested for SARS-CoV-2-RNA by NAAT. SARS-CoV-2-RNA-positive samples were subsequently analyzed by melting curve analysis of NAAT products for the presence of mutations in the viral spike (S) gene (Δ69/70, E484K, N501Y).

Data were collected on standardized case report forms. They were retrospectively and anonymously enlisted in an Excel^®^ database. The collected data included gender, age, reason for presentation, symptoms (dyspnea, fever, cough, rhinitis, sore throat, myalgia, gastrointestinal symptoms, loss of taste, and/or smell), contact with persons tested positive for SARS-CoV-2, SARS-CoV-2 NAAT result, hospital admission, admission to intensive-care units, the presence of chronic diseases (chronic cardiac, oncological, pulmonary, renal diseases, or other chronic disease), results of the mutational analysis, and the Ct value. Due to retrospective nature of the study, and high anonymity Ethics Commission of Cologne University’s Faculty of Medicine waived the need of ethical approval.

In the case of the numerical variables, the descriptive presentation of the results was carried out by specifying the mean value and the standard deviation, or the median indicating the minimum and maximum values in case of not normally distributed variables. We checked for normal distribution using the Kolmogorov−Smirnov-test. Categorical variables were given as a percentage of the underlying collective. Statistical analysis was performed using the Student’s t test for normally distributed values, the Mann−Whitney U-Test for not normally distributed values, or the chi-square test for categorical values. The results were regarded as differing significantly whenever the *p*-value was < 0.05.

## 3. Results

Between 12 January 2021 and 3 June 2021, data from 3544 SARS-CoV-2 NAATs of patients between 0–18 years were collected. Among these, 95 patients tested positive for SARS-CoV-2. For demographic data, see Table 1.

The median age of SARS-CoV-2-positive patients was 9.3 years (min 0.0 years, max 17.8 years), 41.1% (*n* = 39/95) of which reported known contact with a person recently tested positive for SARS-CoV-2 and 55.8% (*n* = 53/95) were symptomatic. They mainly complained about fever, cough, and rhinitis. Of the 95 patients, 14 (14.7%) were admitted to our hospital as a result of COVID-19. There were no fatalities. The median Ct value was 26.0 (min 17; max 41).

In 65/95 (68.4%) cases, an analysis of variants was conducted, and 90.8% (59/65) of these cases carried the typical Alpha-associated constellation of N501Y and DelH69/V70 mutations without an E484K mutation, whereas one patient (1.5%) also tested positive for the E484K mutation. In one patient, we found a mutational pattern consistent with the B.1.351 variant (also known as the Beta variant). In all other patients (4/65), no mutations compared with the previous SARS-CoV-2 wild type variant were found (6.2%). For more data, see Table 2.

The median age of patients with the Alpha mutation pattern was 8.6 years (min 0.0 years, max 17.8 years). Most patients were found in the group from 0 to 12 years of age (71.2%). 45.8% (*n* = 27/59) reported known contact with a person recently tested positive for SARS-CoV-2 and 54.2% (*n* = 32/59) were symptomatic. The most common symptoms were fever, cough, and rhinitis. The median Ct value was 24.0 (min 17.0; max 32.7). For more data, see Table 3.

## 4. Discussion

New mutations of SARS-CoV-2 spread uncertainty worldwide. Not much is known about the clinical course in children infected with the Alpha variant of SARS-CoV-2, which became the dominant variant in Germany between January 2021 and June 2021. Therefore, data concerning the Alpha variant in children are highly needed.

By 3 June 2021, the Alpha variant was found in 90.8% (59/65 patients) of our cases, with Ct values ≤ 33. Data from the German public health institute Robert-Koch-Institut (RKI) show similar numbers. In May 2021, the Alpha variant was detected in over 90% of SARS-CoV-2 cases in Germany [7]. In our clinic, the overall positive rate for SARS-CoV-2 increased significantly in the period of this study compared with data collected in an earlier period of the pandemic (March 2020–June 2020; 2.7% (95/3544) vs. 1.7% (37/2192; *p* = 0.0144)) [10].

The Ct value is of special interest for predicting the potential transmissibility of SARS-CoV-2-positive patients. There are different findings on how a Ct value may reflect the susceptibility of children for SARS-CoV-2, as well as their role in spreading the virus. Ct values in infants (<90 days) are typically found to be very low, indicating a high viral load [11]. On the other hand, most findings of SARS-CoV-2 in children reflect mild symptoms or an asymptomatic course of infection [4], suggesting a lack of causality between viral load and clinical symptoms in the pediatric cohort.

In our cohort of patients with the Alpha variant, the median Ct value was 24.0 (min 17.0; max 32.7). Frampton and colleagues showed that there are significant differences between the Ct values of samples with and without the Alpha variant in adults (mean 28.8 ± 4.7 vs. 32.0 ± 4.8; *p* = 0.0085). They could not attest to a correlation between viral load and mortality [12]. A group from Italy also observed not only significantly lower Ct values in patients infected by the Alpha variant (median Ct value 15.8 vs. other lineages 16.9; *p* < 0.0001), but also a significant longer duration of positivity in nasopharyngeal swabs (median days 16.0 vs. other lineages 14.0; *p* < 0.03) [13]. Clinical significance remains questionable. Surely, there are some noteworthy limitations in general for the determination of the Ct value in direct association with the viral load of a patient. It only reflects an as-is state and the load of viral RNA in a sample can depend on many variables, including sampling and laboratory testing [14]. This can be of special concern in the group of pediatric patients, as deep nasopharyngeal swabs are taken infrequently due to pain and lack of cooperation. Additionally, smaller pediatric swab devices are used, carrying smaller sample volumes [15]. Nevertheless, an association between the viral load and the rapid growth of Alpha variant seems plausible [16].

Over one half of the patients with Alpha variant detection complained of at least one symptom (54.2%) in our study. A recent work from Davies et al. in England suggested that the Alpha variant does not result in a different clinical course compared with the wild type in children [9]. Data about the clinical courses of infections with the Alpha variant in adults are not uniform. Several studies from England and Denmark associated the Alpha variant with an increased risk of hospitalization, as well as a higher mortality in adults [17,18,19]. On the other hand, Graham and colleges found no differences in the course of disease, symptoms, reinfection rate, or transmissibility between Alpha infections and infections without Alpha in an adult English cohort [20].

Looking at the particular symptoms of children in our cohort, patients infected with the Alpha variant mainly complained of fever, cough and/or rhinitis. The Coronavirus (COVID-19) Infection Survey of the UK Office for National Statistics reported that dysgeusia or dysosmia were less common among adult patients with the Alpha variant, compared to patients without detection of the Alpha variant. However, they found cough, sore throat, myalgia and fatigue more often in patients infected with the Alpha variant [21]. Only one of our patients complained about a loss of smell and taste (1.7%). Any conclusion must be drawn with caution, as dysgeusia or dysosmia are symptoms that are hard to objectify, especially in children. Data from the UK looking into clinical characteristics of COVID-19 infections of children admitted to clinical care between January and July 2020 found dyspnea as a common symptom, even before the rising numbers of Alpha; 30% (*n* = 173/570) of their patients showed shortness of breath [22]. In our cohort, we found dyspnea in 8.5% of patients (*n* = 5/59).

In our cohort, 9/59 patients carrying the Alpha variant had to be admitted to hospital (15.3%) and one patient (1.7%) was admitted to the ICU. In an earlier analysis from our center covering March 2020–June 2020, only 8.1% (3/37) of patients tested positive for SARS-CoV-2 had to be admitted to hospital without any necessity to be treated at an ICU (*p* = 0.3598; *p* > 0.9999) [10]. The higher hospital admission rate might be caused by a higher SARS-CoV-2 prevalence in the population per se. The 7-day-incidence in North-Rhine-Westphalia was 6652 cases on 2 April 2020, whereas there was a 7-day-incidence of 34,724 cases on 23 April 2021 [23].

Our data suggest that younger children between the age of 0–12 years are more likely to be tested positive for SARS-CoV-2, because of the occurrence of the Alpha variant (median 8.6 years (min 0.0 years; max 17.8 years)). An epidemiological review comprising data from March–October 2020 in the United States showed that the SARS-CoV-2 incidence was two times higher in adolescents (12–17 years) than in younger children (5–11 years) [24]. This period does not include the Alpha lineage with high certainty, even though testing for Alpha had not been established then. Earlier data from our clinical infection disease center from 2020 also showed a higher median age of 12.0 years (range 0.8–17.4) in SARS-CoV-2-positive tested patients [25]. Potential reasons, besides the higher infectiousness of the Alpha variant [26], may be the establishment of screening patterns in daycare facilities and schools. Daycare facilities and kindergartens stayed open during the recent lockdown in 2021, whereas schools were closed in Germany. Therefore, children between the age of 0–12 years may have had a higher risk of infection than older schoolchildren, who were looked after at home.

The effect of the Alpha variant on children needs to be further determined, especially given the fact that there is no vaccination for those below the age of 12 years available so far, and lockdown-procedures in Germany have included the intermittent closing of schools and childcare facilities for over a year now. The long-term outcomes for children concerning education and psychosocial aspects have yet to surface. However, already today, the disadvantages for children during this pandemic are huge, given that the interests of children are easily neglected in comparison with the economic interests [27,28].

The main limitation of this work is its retrospective and descriptive character. because of the low number of positive NAATs with a proven wild-type virus, statistics on clinical courses should be taken with caution. Nevertheless, we feel this work presents a valuable contribution to the body of knowledge with its comprehensive presentation of real-world clinical data.

## 5. Conclusions

These data provide an overview of the demographic and clinical data of children infected with the Alpha variant in the area of Cologne, Germany. In our cohort, the Alpha variant was responsible for over 90% of cases of infection with SARS-CoV-2 in the period from January 2021 to June 2021. Even though we observed a significant increase of positive rate of SARS-CoV-2 infections in comparison to earlier analyses in the same area, we found no evidence that children infected with the Alpha lineage show more severe symptoms or suffer from a more severe clinical course than those infected with the wild type. The presented results contribute to the comprehensive consideration of future strategies to control this pandemic, and suggest that the emergence of the Alpha variant should not lead to further restrictions to the daily life of children.

## Figures and Tables

**Figure 1 viruses-13-01600-f001:**
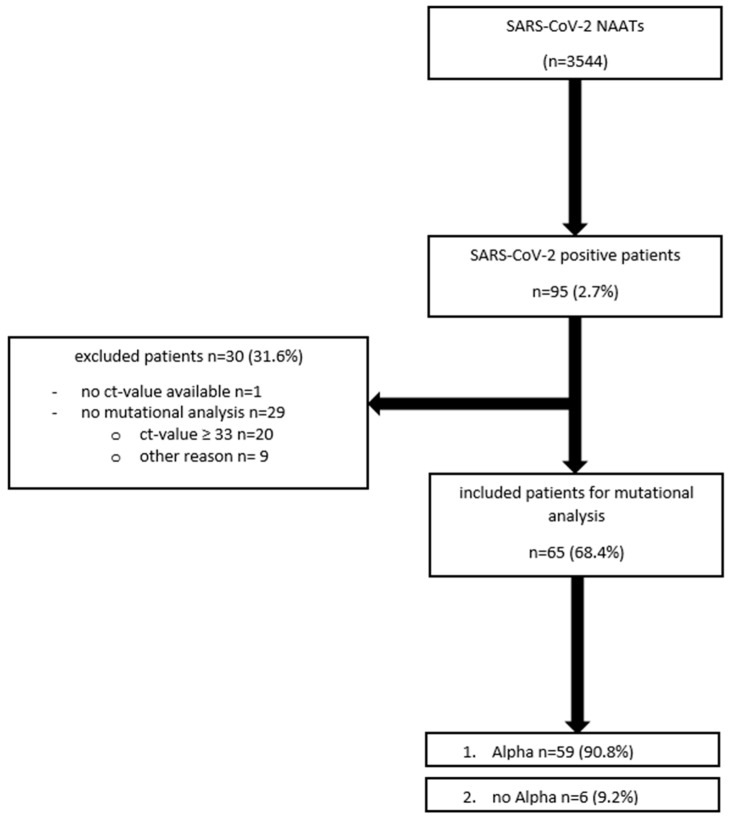
Distribution of patient population (*n* = 3544 nucleic acid amplification tests (NAATs); 12 January 2021–3 June 2021).

**Table 1 viruses-13-01600-t001:** Demographic data on SARS-CoV-2 NAATs (*n* = 3544).

Variable	SARS-CoV-2 NAATs (*n* = 3544)
Age (median)	5.1 years
	(min 0 years; max 17.9)
Gender (male)	53.5% (*n* = 1897)
SARS-CoV-2-Positive NAATs	2.7% (*n* = 95)
Admission to Hospital	27.5% (*n* = 976)
Underlying Chronic Disease	41.8% (*n* = 1480)
Contact to Somebody Recently Tested Positive for SARS-CoV-2	5.6% (*n* = 199)
Symptomatic	19.8% (*n* = 703)
Mortality	0.0% (*n* = 0)

NAAT—nucleic acid amplification test.

**Table 2 viruses-13-01600-t002:** Patients tested negative for SARS-CoV-2 vs. patients tested positive (Alpha mutation pattern: N501Y, Del69/70: negative for E484K; Alpha mutation pattern with escape mutation: N501Y, Del69/70: positive for E484K; Beta mutation pattern: N501Y, E484K, and K417N; negative for Del69/70).

Variable	Negative SARS-CoV-2 NAATs (*n* = 3449)	Positive SARS-CoV-2 NAATs (*n* = 95)	*p*-Value
Age (median)	5.0 years	9.3 years	*p* < 0.0001
	(min 0.0 years; max 17.0 years)	(min 0.0 years; max 17.8 years)	
0 ≤ 3 years	37.2% (*n* = 1284)	21.1% (*n* = 20)	*p* = 0.0011
3 ≤ 6 years	19.2% (*n* = 663)	10.5% (*n* = 10)	*p* = 0.0333
6 ≤ 12 years	20.4% (*n* = 705)	31.6% (*n* = 30)	*p* = 0.0014
12 ≤ 18 years	23.1% (*n* = 797)	36.8% (*n* = 35)	*p* = 0.0030
Gender (male)	53.6% (*n* = 1848)	51.6% (*n* = 46)	*p* = 0.3486
SARS-CoV-2-Positive NAATs	-	100.0% (*n* = 95)	-
Suffering from a	42.6% (*n* = 1470)	10.5% (*n* = 10)	*p* < 0.0001
Chronic Disease			
Contact to Somebody Recently Tested Positive for SARS-CoV-2	4.6% (*n* = 160)	41.1% (*n* = 39)	*p* < 0.0001
Symptomatic	18.8% (*n* = 650)	55.8% (*n* = 53)	*p* < 0.0001
Fever	0.0% (*n* = 0)	28.4% (*n* = 27)	*p* < 0.0001
Cough	5.9% (*n* = 203)	27.4% (*n* = 26)	*p* < 0.0001
Rhinitis	6.1% (*n* = 210)	25.3% (*n* = 24)	*p* < 0.0001
Headache	2.9% (*n* = 100)	9.5% (*n* = 9)	*p* = 0.0023
Dyspnea	1.3% (*n* = 46)	4.2% (*n* = 4)	*p* = 0.0436
Sore Throat	2.6% (*n* = 88)	4.2% (*n* = 4)	*p* = 0.3115
Gastrointestinal Symptoms	4.1% (*n* = 142)	8.4% (*n* = 8)	*p* = 0.0617
Myalgia	0.2% (*n* = 7)	3.2% (*n* = 3)	*p* = 0.0020
Dysgeusia and Dysosmia	0.2% (*n* = 6)	7.4% (*n* = 7)	*p* < 0.0001
Admission to Hospital	27.7% (*n* = 955)	22.1% (*n* = 21)	*p* = 0.2464
Ct Value (median)	-	26.0 (min 17.0; max 41.0)	-
Variant Analysis	-	68.4% (*n* = 65)	-
Wild Type	-	4.2% (*n* = 4)	-
Alpha Mutation Pattern	-	90.8% (*n* = 59)	-
Alpha Mutation Pattern with Escape Mutation	-	1.5% (*n* = 1)	
Beta Mutation Pattern		1.5% (*n* = 1)	-

**Table 3 viruses-13-01600-t003:** Patients tested positive for the Alpha variant (*n* = 59) vs. patients tested negative for the Alpha variant (*n* = 6).

Variable	Alpha Variant (*n* = 59)	No Alpha Variant (*n* = 6)	*p*-Value
Age (median)	8.6 years	14.0 years	*p* = 0.0953
	(min 0.0 years; max 17.8 years)	(min 7.1 years; max 16.3 years)	
0 ≤ 3 years	20.3% (*n* = 12)	0% (*n* = 0)	*p* = 0.5831
3 ≤ 6 years	13.6% (*n* = 8)	0% (*n* = 0)	*p* > 0.9999
6 ≤ 12 years	37.3% (*n* = 22)	44.4% (*n* = 2)	*p* > 0.9999
12 ≤ 18 years	28.8% (*n* = 17)	66.6% (*n* = 4)	*p* = 0.0800
Gender (male)	55.9% (*n* = 33)	44.4% (*n* = 2)	*p* = 0.4025
Suffering from a	3.4% (*n* = 2)	0% (*n* = 0)	*p* > 0.9999
Chronic Disease			
Contact to Somebody Recently Tested Positive for SARS-CoV-2	45.8% (*n* = 27)	16.7% (*n* = 1)	*p* = 0.2245
Symptomatic	54.2% (*n* = 32)	66.6% (*n* = 4)	*p* = 0.6841
Fever	25.7% (*n* = 15)	0.0% (*n* = 0)	*p* = 0.3225
Cough	23.7% (*n* = 14)	44.4% (*n* = 2)	*p* = 0.6306
Rhinitis	23.7% (*n* = 14)	44.4% (*n* = 2)	*p* = 0.6306
Headache	11.9% (*n* = 7)	2.9% (*n* = 1)	*p* = 0.5607
Dyspnea	8.5% (*n* = 5)	0.0% (*n* = 0)	*p* > 0.9999
Sore Throat	6.8% (*n* = 4)	44.4% (*n* = 2)	*p* = 0.0908
Gastrointestinal Symptoms	6.8% (*n* = 4)	0.0% (*n* = 0)	*p* > 0.9999
Myalgia	5.1% (*n* = 3)	0.0% (*n* = 0)	*p* > 0.9999
Dysgeusia and Dysosmia	1.7% (*n* = 1)	16.7% (*n* = 1)	*p* > 0.9999
Admission to Hospital	20.3% (*n* = 12)	0.0% (*n* = 0)	*p* = 0.5831
for COVID-19	15.3% (*n* = 9)	0.0% (*n* = 0)	*p* = 0.5838
Ct Value (median)	24.0 (min 17.0; max 32.7)	25.0 (min 19.0; max 32.0)	*p* = 0.3886

## Data Availability

Data available on request due to restrictions e.g., privacy or ethical.

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
