# Peer review of "The Alpha Variant (B.1.1.7) of SARS-CoV-2 in Children: First Experience from 3544 Nucleic Acid Amplification Tests in a Cohort of Children in Germany"

_viruses, 2021, doi:10.3390/v13081600_

Round 1
Reviewer 1 Report
This paper provides information on the Alpha variant (B.1.1.7) of SARS-CoV-2 in children and adolescents, age 0 to 19 years old, in Germany. In particular, information on the prevalence and symptoms of the disease among individuals in this age group is provided. Details of this nature are quite interesting due to the paucity of such information in existing medical research. In addition, this paper likely will motivate additional and more thorough research on this topic in subsequent analyses and additional countries. As such, the paper addresses a useful and interesting topic.
I offer the following observations as a mean for improving the manuscript.
- The main shortcoming of this paper is the lack of statistical evidence to support the claims made by the authors. For example, the authors state that they find:
“. . . no evidence that 29 children infected with Alpha lineage show more severe symptoms or suffer from a more severe 30 clinical course than those infected with the wild type”,
and
“. . . no evidence that children infected with Alpha lineage show more severe symptoms or suffer from a more severe clinical course than those infected with the wild type.”
and
“We can confirm early findings that children are less affected by SARS-CoV-2 infection even with the spreading of the Alpha lineage.”
However, no rigorous statistical tests are provided to back up these assertions. Basic summary statistics are provided, and it seems that inference is drawn based on “eye balling” these statistics. It would be more convincing to see tests performed on these statistics—at the very minimum, an equality of means test between the comparison groups.
- The paper requires significant editing of English grammar, vocabulary and clarification throughout. Just to provide one example: “June 3th” should be written as “June 3rd”. In addition, for purposes of clarification, in the first sentence of the abstract, instead of referring to “. . . the Alpha variant . . .”, I suggest you refer to: “. . . the Alpha variant (B.1.1.7) of SARS-CoV-2 . . .”. This is good practice for the first time the variant is mentioned. Furthermore, I suggest the authors adopt one consistent method for using a period when citing numerical values (and stop using it for any other purpose). That is to say, in some places in the manuscript, the period is used to indicate a decimal place; in other places, the period is used to separate three whole numbers. This is confusing and frustrating to the reader. I suggest the use of a decimal exclusively to indicate a decimal place, and the use of a comma to separate (groups of three) whole numbers.
Reviewer 2 Report
GENERAL COMMENTS: The manuscript provides a general description of the prevalence of the alpha variant in a single center in Cologne, Germany. It is descriptive in nature only, with little analysis into the nature of the exposure which may have led to the infections that were discovered. In addition, it provides little data with regards to the outcomes of the infected children, and little comparison - other than brief demographics - to the noninfected members of the cohort.
ABSTRACT: No specific recommendations
INTRODUCTION: It could be shortened, as it contains information about the alpha variant that may not be pertinent to the article other than a review of the variant itself, which is somewhat off-topic.
I am not certain that the penultimate paragraph of the introduction is necessary. At the least, it should be moved to the discussion. The manuscript itself does little to address the concerns conveyed here.
MATERIALS and METHODS: No specific recommendations.
RESULTS: Tables 1, 2 and 3 should be combined into one table with separate columns for all members of the cohort, SARS-CoV-2+, and alpha+. In addition, it would be helpful to see the symptoms in the all-members column. Presumably, many patients who tested negative had similar complaints as those who tested positive. Basic descriptive statistical analysis comparing the groups would be helpful.
Pie charts are generally unnecessary, and similar data could be included as rows under the overall age with appropriate percentages.
DISCUSSION: No specific comments, though the above suggestions may make the discussion easier to interpret.
CONCLUSION: No specific comments.
